# Complexities of Intergovernmental Relations in Water Service Provision: A Developmental Local Government Perspective

**Avhavhudzani Khangale [1], Onkgopotse Senatla Madumo [2,*] and Michel Mudikolele Tshiyoyo [1]**

[1] School of Public Management and Administration, University of Pretoria, Hatfield 0083, South Africa; avhavhudzani@gmail.com (A.K.); michel.tshiyoyo@up.ac.za (M.M.T.)

[2] School of Public Management, Governance and Public Policy, University of Johannesburg, Auckland Park 2006, South Africa

* Correspondence: onkgopotsem@uj.ac.za

**Abstract:** Water is a fundamental human right, and its provision is essential for the maintenance of the general quality of life. The South African government has a constitutional obligation to provide clean potable water to all citizens. This study explores the practice of cooperative government among the three spheres of government in the provision of water services to communities in South Africa. It also seeks to analyse the application of the principle of cooperative government as an effective tool for ensuring water service delivery in local government. Thus, to achieve these objectives, the study addresses two research questions: first, what are the complexities associated with intergovernmental relations in the efforts to provide water services to communities? And secondly, how could cooperative government be utilised as a mechanism for water service delivery in local government? This article begins by examining the literature on intergovernmental relations and cooperative government to provide a comprehensive understanding of the topic. A qualitative research method is applied, where interviews were conducted to determine the impact of cooperative governance on water governance. This study sheds light on the complexities associated with intergovernmental relations in the provision of water services to communities. The findings of this study recognise the need for municipalities to continuously monitor and improve their water service delivery strategies and water service delivery plans, to align with the conditions of the municipality and needs of the people. This is significant as it provides useful insights to policy makers, water service providers, and researchers in the field of intergovernmental relations and water service provision on how to address challenges associated with water service delivery within the intergovernmental relations context.

**Keywords:** water services; intergovernmental relations; service delivery; cooperative government

## 1. Introduction

Access to basic water service and basic sanitation service has been acknowledged as fundamental human rights. Along with the realisation of basic human rights, achieving equity in water and sanitation services is the main target of Sustainable Development Goal (SDG) 6 ("Ensure availability and sustainable management of water and sanitation for all"; 6.1 and 6.2, respectively). The equitable allocation of both basic water and basic sanitation services is deeply embedded within the objective of the social system of sustainable development (Duran et al. 2015), and is indirectly addressed in SDG 10 ("Reduce inequality within and among countries"; 10.2, 10.3, and 10.B, respectively). (Bayu et al. 2020, p. 1).

Potable water supply is a fundamental basic human right protected by international conventions and national laws (WHO 2014). For communities to lead a healthy, productive, and dignified lifestyle, access to an adequate supply of potable water is indispensable (Haylamicheal and Moges 2012; WHO 2014). In addition, the reality is that approximately 884 million people in the world still use unprotected potable water sources like springs, fountains, wells and ponds that are open to contamination, leading to waterborne diseases (Malima et al. 2021, p. 169).

Although the provision of water and sanitation is a priority for many countries, particularly lower-income countries, in South Africa, it is considered to be a basic human right supported by the supreme law of the country for various reasons but mainly because of the consequences of the imbalances experienced in the past. To redress the legacy of inequalities in the provision of water and sanitation, it is imperative for government authorities to devise appropriate means to prioritise the delivery of clean potable water and sanitation as widely as possible. The need being pressing at local government sphere where indigent and vulnerable communities are located, it is essential that effective intergovernmental relations occur between the three spheres of government (i.e., national, provincial and local) so they can collaborate to ensure that water as an essential resources, is extracted, transferred to appropriate plants to be treated before it can be distributed to communities in the safest way possible. The advent of democracy in 1994 created an environment for the running of a credible system of cooperative government and intergovernmental relations. For instance, the Constitution of the Republic of South Africa, 1996 (South Africa 1996) provides for the establishment of an appropriate mechanism to encourage collaboration among the different government departments, entities, and spheres of government. Beyond collaboration and interaction between government department and authorities, there is also provision for interaction with other role-players such as the private and the non-profit sectors. This article aims to explore the practice of cooperative government among the three spheres of government in the provision of water services to communities in South Africa with a specific reference to the Vhembe District Municipality (VDM). The key questions that this article intends to answer are as follows: (i) What is the nature of the practice of cooperative government, among the three spheres of government, in the process of providing water service to communities? (ii) to what extent can the principle of cooperative government be considered as an effective tool for water service delivery in a district municipality? In the process of answering these questions this article examines the challenges facing cooperative government particularly in the provision of water and sanitation at the VDM. This article is composed of seven sections. Section 1 will address the methodological component of this article. Section 2 discusses the theory that is associated to intergovernmental relations and cooperative governance. Thirdly, the policy framework that gives impetus to the provision of water services in South Africa is highlighted. Section 4 elaborates on the institutional arrangements for water service delivery. Section 5 will discuss the critical requirements for the existence of intergovernmental cooperation in the delivery of water services. Section 6 highlights the current situation of water service availability. Section 7 is dedicated to discussion and presentation of findings before this article can conclude and provide some recommendations.

## 2. Methodology

The purpose of this research is to explore the challenges associated with cooperative government and intergovernmental relations in the quest to ensure the provision of safe and reliable water in a particular district municipality in South Africa. This study was undertaken at the VDM, which is one of the five district municipalities in the Limpopo Province, South Africa. District municipalities are often established in rural areas to provide administrative, fiscal, and infrastructural support to the individual local municipalities (Madumo and Koma 2019, p. 583). A qualitative research method was used in the study on which this article is based. Specifically, the case study research design was utilised to explore the challenges faced by the VDM in providing water services within the system of intergovernmental relations. When using a case study design, the research obtains extensive data on the individuals or circumstances on which the investigation is focused (Leedy and Ormrod 2014, p. 143). Thus, this type of research emphasises personal knowledge and conveys the individual opinions of participants in the research. We used purposive sampling to select the experts working within the various water service value chain within the VDM. As a result, these experts are under the employ of the VDM within the line functions of technical water services. The experts that participated in this study included

individuals occupying the following positions, manager in technical services, manager in operations and maintenance section, director in the project management unit, assistant director in planning and water provision unit, director in water demand section, workers in the water quality services unit. These experts are key in ensuring coordination of the delivery of water services within the district municipality. The water experts should have knowledge and experience on the processes for facilitating the delivery of water services, policies, and framework for cooperative government in relation to water service governance. We targeted to interview at least fifty percent of the identified experts within VDM. Data were collected using semi-structured and face-to-face interviews. The semi-structured interviews were relied on to probe the participants on the challenges relating to cooperative government within the provision of water service delivery in Vhembe. The interview questions were posed in such a way that addresses the mechanisms, processes and coordinating efforts of ensuring water service delivery. The type of the interview questions asked focused on two categories, the first set of questions aimed at understanding the various aspects associated with water service delivery from a perspective of a municipality. The second set of questions sought to probe the challenges associated with cooperative governance within the context of a municipality.

Since the interview process is flexible, we were able to probe the participants where necessary, in order to develop a better understanding of phenomenon we were investigating (Bryman 2012, p. 471). We used secondary data, including the review of literature, reports, and documents from the VDM to build up an argument.

### 3. The Emergence of Intergovernmental Relations and Cooperative Governance

The New Public Management (NPM) gained prominence in the early 1990s and its precursors advocated for principles that aimed at improving productivity and performance of the public sector in the governments across the world (Fatemi and Behmanesh 2012). Thus, both intergovernmental relations and cooperative governance were intensified and became means that were prioritised and used to achieve mandates of the various governments. This usage was accelerated by the conditionalities on loan agreements that were set out by the United Nations through the Bretton Woods institutions, i.e., International Monetary Fund and the World Bank to countries that received loans from them (Easterly 2005). The system of intergovernmental relations contributes to the establishment and strengthening of the principles of cooperation among the three spheres of government. Thus, the different spheres of government, exist to promote better collaboration as espoused by democracy and the South African constitution. Cooperative government requires the government to have the ability to recognise its multifaceted nature and ensure proper mechanisms to provide support across the different spheres of government (Madumo and Koma 2019, p. 582).

### 3.1. Institutional Theory

Many governance arrangements involve spatial units with highly unequal powers. Whether a feudal monarchy and its principalities, a hegemon and its peripheral units, a national government and its subnational entities, or a regional government and its local entities, a pivotal question of institutional design is how much authority the dominant unit (A) cedes to the subordinate Unit (B). It is in this context that one can stress that a 'direct' style rule features a more decentralised framework in which important decision-making powers are delegated to the weaker entity (Gerring et al. 2011, p. 377). In this era of governance, it has become evident that governments around the world are striving to use various means to provide services to citizens, and that service delivery is no longer the sole mandate of governments. In addition to the government, many other role-players are brought in so that they can contribute to the delivery of services, and South Africa is not an exception. As already stressed above, South Africa has three spheres of government. When it comes to the provision of water services, the Department of Water and Sanitation (DWS) is the custodian of the country's water resources, it is mandated to ensure effective supply

of water and sanitation services to all South African citizens. Several pieces of legislation such as the National Water Act (NWA), Water Services Act (WSA) and Water Research Act (WRA) support the spirit of the Constitution and entrust the DWS with the responsibility to ensure that the population of the country has clean water. In achieving this mandate, DWS relies on its internal and external stakeholders. Local government is one of the stakeholders since it is a sphere that is closer to the grassroots.

Institutional Theory considers institutions as being multifaceted, durable social structures, made up of symbolic elements, social activities, and material resources. This theory relies on a collection of ideas and mechanisms upon which social structures and behaviours are supported and restricted (Bjorck 2004; Scott 2013). Furthermore, Scott (2013) identified three pillars of institutions including regulative, normative, and cultural-cognitive pillars. The regulative pillar underscores the importance of regulatory processes, rule-settings, monitoring, and sanctioning activities regarding how institutions work and function. The normative pillar emphasises social obligation and normative rules towards adopting new structures. The third pillar is the cultural-cognitive, which focuses on the role of cultural-cognitive elements of institutions and the shared values that represent the nature of social reality and how meanings are constructed and created. For the sake of this article, the regulatory pillar is relevant as South Africa has numerous pieces of legislation as well as regulatory frameworks that guide the provision of water services and determine which role-players is responsible for which services. The national government has a specific mandate whereas the provincial and local government along with private actors have specific mandates and are all working towards achieving the national vision in terms of water and sanitation.

### 3.2. Fostering Intergovernmental Relations through a Cooperative Government

In the context of South Africa, intergovernmental relations refer to a framework that encourages the different spheres of government within the country interact, cooperate, and coordinate their efforts to address common policy objectives to deliver public services seamlessly. The system of cooperative government recognises three spheres of government, i.e., national, provincial, and local. Thus, intergovernmental relations aim to establish mechanisms to enhance collaboration, consultation, and promote maximum coordination among these spheres to ensure effective governance and service provision. Chapter 3 of the Constitution of the Republic of South Africa, 1996 provides guidelines for the establishment of the principles of cooperation.

The enactment of the Intergovernmental Relations Framework Act, 2005 (Act 13 of 2005), introduced a legislative framework to promote collaboration between and among the various spheres of government, thereby further establishing mechanisms and procedures to facilitate the settlement of disputes. Through the intergovernmental relations framework a platform is established for the spheres of government to collaborate on governance matters, including; policy development, resource allocation and utilisation, planning and implementation, and the coordination of service delivery initiatives.

Municipalities are considered the basic governance units of local government in South Africa. There are different categories of municipalities, namely: metropolitan, district and local municipalities (Thornhill 2012, p. 134). Since the focus of our research is predominantly in a rural area, thus district and local municipalities are the focus of this article. A district municipality is a municipality that has executive and legislative authority in an area that contains more than one local municipality (Section 155 of the Constitution of the Republic of South Africa). In other words, a district municipality is made up of several local municipalities. As such the Vhembe District Municipality is comprised of four local municipalities, namely: Collins Chabane, Makhado, Musina and Thulamela local municipalities. In accordance with Section 84 of the Local Government: Municipal Structures Act, 117 of 1998, district municipalities are water service authorities (as defined in the Water Services Act, 107 of 1997) which are mandated to deliver bulk water supply to the communities and households within their areas of jurisdiction. Despite this clear

mandate, Clifford-Holmes (2015) alludes that most municipalities that face water service delivery challenges are in larger towns and rural areas.

South Africa adopted a democratic model of cooperative government among the three spheres of government. Section 40 of the Constitution, 1996 emphasises that cooperative government should be fostered among the national, provincial, and local spheres of government which are interconnected, distinct, and inter-dependent. The main idea is to decentralise the delivery of services through the system of subsidiarity and consequently bringing government closer to the people and address the challenges of communities (Buire 2011). The interaction, collaboration, and support of the three spheres of government, as outlined in the Constitution, is a fundamental aspect of cooperative government. Cooperative government therefore plays a significant role, particularly where national and provincial programmes require implementation at the local sphere. For intergovernmental relations to succeed, each sphere of government should recognise its role and account for its responsibilities within the cooperative government value chain.

### 3.3. The Role of the National Government in Cooperative Government

The national sphere of government is accountable for the development of the policies or frameworks that guide basic service delivery. The water service infrastructure framework, for example, is developed at national level for implementation by district municipalities (DCOG 2019). Cooperative government ensures that the integrity of each sphere of government is maintained, and it should be used to streamline the delivery of fundamental services from the national to the local spheres. The national government contributes to the function and duties of municipalities and plays a strategic role by providing an overall legislative framework for local administration. Since the Constitution, 1996 has listed the provision of Water as a basic human right, the Department of Cooperative Governance (DCOG), therefore becomes responsible in ensuring that district municipalities undertake their responsibility to make this a reality. The Department of Cooperative Governance, as custodian of local government is also responsible for providing constant support to municipalities in the effort to enable them to deliver basic services, thereby, putting people and their concerns first, and promoting and strengthening cooperative governance (DCOG 2019).

### 3.4. The Role of Provincial Government in Cooperative Government

The provincial sphere of government has the primary responsibility of basic service delivery through the implementation of programmes that aim to deliver a wide range of services to citizens. Malan (2005) alludes that provincial government should promote local government capacity development for municipalities to be in position to carry out their roles and function and administer the matters of their jurisdictions. Asha and Makalela (2020) highlights that provincial government should support municipal capacity to strengthen and improve basic service delivery to communities. In accordance with Section 154 of the Constitution, 1996, provincial sphere of government should develop a mechanism to identify challenges that municipalities in their jurisdiction and subsequently provide support to facilitate effective provision of services. Thus, cooperative government could be used as a tool to improve municipal capacity and enhance basic service delivery. The provincial government deals with issues within the jurisdiction of the provinces (DCOG 2019), and derives some of their responsibilities from the functions listed in Schedule 5 of the Constitution, 1996.

### 3.5. The Role of Municipalities in Cooperative Government

As a sphere closest to the people, local government is obligated to deliver basic services such as water, sanitation, electricity and refuse removal to communities through their respective municipalities. Malan (2005) argues that the integration of local government into the system of cooperative government has demonstrated a multifarious system but created an innovative opportunity for a more receptive and efficient government. Van

der Linde (2006) indicates that the DWS plays the role of sector leader in ensuring that local government institutions provide the support promulgated in Section 154(1) of the Constitution of 1996. Some of the problems within the water sector could be attributed to the coordination challenges between the water agencies and municipalities (Araral and Wang 2013). For example, in Limpopo Province, Sekhukhune District Municipality has experienced interruptions and shortage in water provision due to the lack of proper coordination between the provincial government, municipalities and water agencies or boards. Cooperative government could be used to address some of the problems facing the administration in the distribution of basic services, particularly to the municipalities. Weaver et al. (2017) identify a strong relationship between the existence of municipalities and water service provision. This is attributed to the notion that basic service delivery, i.e., water provision, lies at the centre of the responsibilities of municipalities.

In the light of the above, it is important to note that, in South Africa, the institutional theory is effectively applied as there are indications on the respective mandates of each sphere of government as well as the nature of contributions made by private actors towards the attainment of the national vision particularly in terms of the provision of water and sanitation. In this context, the application of the principle of cooperative government becomes paramount to the realisation of the country's vision in this sector.

## 4. Policy and Legislative Framework for the Provision of Water Services in South Africa

The Department of Water and Sanitation (2010) acknowledges the two pieces of legislation that drive the national strategic, water governance and regulatory frameworks as the National Water Act (Act 36 of 1998) and the Water Service Act (Act 108 of 1997). These Acts govern water use and water-resource management in South Africa. In addition, the National Environment Management Act (Act 107 of 1998) contributes to the legal framework of water regulation by providing principles for decision making on environmental issues. It is important to note that there is a strong correlation between Chapters 2 (Bill of Rights) and 3 (cooperative government) of the Constitution (South Africa 1996), which is critical for sustainable and effective delivery of basic services. The following legislation contributes to water service in South Africa.

### 4.1. Constitution of the Republic of South Africa, 1996

The Republic of South Africa is a country founded on democratic principles with the Constitution, 1996 as a basis of democracy and establishes the three spheres of government. These spheres of government are vested with legislative and executive functions and powers (South Africa 1996). The Bill of Rights as promulgated in the Constitution, 1996 indicates the distribution of basic services as processes to enhance the standard of living of poverty-burdened citizens. The Constitution assigns the responsibility of water service delivery to the district municipalities. Section 156(1) of the Constitution makes a provision for district municipalities to have an executive authority and the right to somewhat manage water facilities (i.e., water supply systems and reservoirs) in efforts to provide access to clean drinkable water. Based on this explanation, VDM is a water service authority that is constitutionally obligated to ensure that citizens receive access to sustainable water services (Vhembe IDP 2020).

### 4.2. Strategic Framework for Water Services

Strategic framework is defined by the Department of Water Affairs (2003) as a systematic approach to water service provision in South Africa. It outlines water service provision ranging from small community schemes supplying water in isolated rural areas to regional schemes supplying water to larger urban areas (Department of Water Affairs 2003). In addition, the framework emphasises that the DWS should not be operational in terms of service provision but should support and regulate those institutions that provide water services to the people. The Strategic Framework for Water Services (SFWS) addresses the whole range of sanitation and water supply matters, serving as a comprehensive framework for

the water services sector in its entirety (Department of Water Affairs 2003). It provides the overall objectives of integrated water-resource management, and defines the institutional and operational structure (finance, planning and implementation) that need to be in place to attain these goals.

### 4.3. Water Services Act, 1997

The Water Services Act (Act 108 of 1997) is one of the principal legislations in South Africa's water services sector, mainly in the district municipalities, that supports the Bill of Rights through the regulation of water and sanitation services are regulated to ensure that all people are entitled to a minimal level of service. To ensure that water provision is consistently administered, Section 9 of the legislation permits the minister to recommend mandatory national norms and standards for water services delivery.

In terms of water usage, the Water Services Act, 1997 indicates that everyone is entitled to basic water provision, and any water service agency should take appropriate efforts to ensure that the right to water is realised. The legislation further states that the local and district municipalities are the water service authorities in their areas of jurisdiction and may themselves provide the water services or contract a service provider that specialises in water provision to do so.

### 4.4. National Water Act, 1998

At the national level, the National Water Act (36 of 1998) acknowledges the scarcity of water and unequal distribution of resources in the country that resulted from the biased laws and practices of the apartheid regime and appreciates that water is a basic need for all people. In addition, the legislation is the acknowledgment of the national government that it has overall responsibility for and authority over water resource management, including water distribution and matters that affect international waters, and recognition of the importance of water use to benefit people.

## 5. The Institutional Arrangement of Water Service Delivery in South Africa

In South Africa, the institutional landscape of policy and legislation has changed tremendously since 1995. Madigele (2017) emphasises that the general review of water laws resulted in the formation of the White Paper on National Water Policy (South Africa 1997), which was accompanied by the enactment of the National Water Act, 1998 (Act 36 of 1998; South Africa 1998), which focuses on the decentralised model of governance to redress discrepancies in the water services sector.

Schreiner et al. (2011) point out that South Africa's institutional arrangement for the regulation of water resources is relatively multifaceted owing to the various role players, including Parliament, the Water Tribunal, water services authorities and courts. Parliament is responsible for establishing and amending legislation that regulates the use of water and ensuring the effectiveness for DWS. In South Africa, the institutional arrangement of water services remains a challenge that impacts the distribution for basic service to local communities (Weaver et al. 2017).

In the water sector, the government has forged a partnership with NGO's water users and the private sector as a means of regulatory control in the use of water service. The Water Research Commission (2016, p. 16) indicates that the DWS is responsible for regulating the use of water by private users and district municipalities are included in the state organ and other sectoral departments. The regulation of district municipalities in the use of water in South Africa is a responsibility of the local government, which is supported and regulated by the DCOG.

The institutional arrangement for the delivery of water in the district has seen a rise in the number of water concerns and as a result there is a need to develop an appropriate legal framework that can address the delivery of water services (Lawyers for Human Rights 2009). It is essential that the local sphere comprehend their constitutional obligations in how they operate with regard to water service delivery.

The Water Service Act, 1997 (Act 108 of 1997) provides for the establishment of an institutional framework to ensure the delivery of water services to communities. As per the prescribed framework, various institutions have been set up to oversee and control the water provision, and these vary from one district municipality to another.

These institutions comprised of water service authorities; water services providers with which district municipalities have entered into agreements for the supply of water; water boards that play a role in the supply of bulk water to other water services institutions in a specified geographic area; water services committees; and water services intermediaries that ensures the quantity, qualities and sustainability's of water meet the prescribed minimal standards.

## 6. The Requirements for Intergovernmental Cooperation in the Delivery of Water Services

In the previous sections, light was shed on the legal framework and institutional arrangements that guides the provision of water services in South Africa. Now, this section examines the provisions made in terms of the promotion of cooperation between the three spheres of government for effective water services delivery. In South Africa, there are various pieces of legislation and by-laws that address basic water service delivery at the level of district municipalities. Some of legislation are guided by the Constitution, while others are developed and implemented by the district municipalities, such as by-laws. There are various by-laws adopted by respective municipalities which intend to encourage compliance and alignment with legislation enacted by the provincial and national governments. The following requirements are among the key that contribute to cooperative government and the delivery of water service in the district municipalities.

### 6.1. Principles, Mechanisms and Procedures

The Local Government: Municipal Systems Act, 2000 (Act 32 of 2000) establishes the principles, mechanisms, and procedures that are necessary for district municipality to uplift local communities by providing basic services including water, and to ensure that all people have access to services that are affordable. Furthermore, Section 78 of the Municipal Systems Act, which highlights that in their review of delivery mechanisms, municipalities must first assess whether the service can be provided through an internal mechanism, after which they can explore external mechanisms as a means of delivery. In addition, the legislation defines how the local sphere should function and what partnership agreements district municipalities may conclude, to ensure the distribution of water to the local communities. Thus, Tissington (2011, p. 6) points out the emphasis by the Municipal Systems Act, that municipalities are required to develop an indigent policy to assist poor households in accessing basic municipal services, including safe drinking water.

### 6.2. Operational Requirements for Service Delivery

The Local Government: Municipal Structures Act, 1998 (Act 117 of 1998) establishes the framework for the development of municipalities in the numerous categories defined by the Constitution, namely district, local and metropolitan municipalities. The legislation prescribes the functions and operational requirements of the municipal councils, and the internal structure and functions of the municipalities. Kraai et al. (2017) allude that the Municipal Structures Act, 1998 outlines the legal nature of the municipalities, which form part of the system of cooperative government, and elucidates the obligations for the municipal administration and council as well as the rights of the local communities.

## 7. Current State of Affairs

Owusu-Ampomah and Hemson (2004) describe service delivery as playing a greater role in local government in South Africa and other developing countries than in developed countries. Because of constitutional provisions and high poverty levels, they argue, service delivery in South Africa is seen as an instrument and social contract to create social inclusion

and raise living standards of the poor majority previously excluded by the apartheid government. The prevalence of the principle of cooperative government in the delivery of public services in South Africa requires a proper coordination and facilitation of the process between the various role-players involved the provision of such services.

According to Edokpayi et al. (2018) almost 2.1 million people in South Africa lack access to safe water infrastructure. Thus, an estimated 1.8 billion people globally drank water that is not safe. The developing countries have been the most affected, wherein the compromised hygiene practices and premature death in young children are attributable to the lack of access to clean water. One of the challenges for many municipalities is the provision of basic water services to all South Africans. Water scarcity is one of the world's most pressing issues, and authorities are frequently faced with the challenge of ensuring that adequate water resources are available to meet the ever-increasing demand. Water scarcity is primarily a problem in semi-arid regions, where it is difficult to secure water supplies due to high population density. Several factors, such as settlement location, socio-economic and demographic variables such as household size, and environmental variables such as temperature and precipitation, have been found to influence water consumption at the household level in rural areas (Shan et al. 2015; Murwirapachena 2021; Priyan 2021). Many rural areas in South Africa are becoming increasingly dissatisfied with the quality, quantity, accessibility, and frequency of interruptions in water delivery (Statistics South Africa 2018). Limpopo is a water-scarce province that was declared a disaster area in November 2015 because of the drought that affected most of South Africa's provinces. Water scarcity is a significant development issue in Limpopo Province, limiting both economic and social activities (Mogooe and Muyengwa 2021). Water-related challenges continue to affect communities throughout the province, but the Vhembe District Municipality is at the epicenter of this (Rankoana 2023, p. 251). For instance, Vhembe is one the five district municipalities of the Limpopo province and it is recognised as a rural area which is faced with many challenges like any other municipality falling in this specific category of municipalities.

According to a study by Magidimisha and Chipungu (2019) as far as access to running water is concerned, 90% of households in urban areas indicated that they have access to such a service as compared to only 10% of the respondents in rural areas. Further disaggregation of water provision shows that there are a variety of sources in both rural and urban areas. While 90% of respondents in urban areas indicated that they have direct access to public tap water, other households (10%) also indicated that they obtained water from boreholes, dams and rivers. However, rivers, dams and boreholes are largely used as sources of bulk water supply which in turn feed into public taps. Vhembe District Municipality is one of the least performing district municipalities in South Africa in terms of access to water and sanitation services, with only 7.4% of households having access to piped water inside their dwelling and 16% of the households having access to a flush toilet connected to the sewage system (The Local Government Handbook 2023).

## 8. Discussion and Findings

In the quest to understand the cooperative government challenges in addressing water service delivery in municipalities, interviews were conducted with the officials from the Technical Services Department of the Vhembe District Municipality. This section presents the research findings which were carefully drawn from interviews, literature and data analysis. Twelve experts including managers, technicians and engineers were interviewed. Subsequently, common themes, ideas and patterns of qualitative data were identified to establish a narrative analysis. The reporting of the results is segmented into two parts: the first one discusses aspects relating to water service delivery, whereas the second one examines aspects that relate to cooperative governance.



*8.1. Water Services Delivery*

This section presents findings addressing the objective that aim at exploring water service delivery in Vhembe District Municipality. Consequently, results of the interviews have been summarised as follows:

*8.2. Types of Water Resources Available to Ensure Adequate Supply in the Jurisdiction of the Municipality*

In the quest to understand the types of water resources available in the Vhembe district, some research participants indicated that available resources differed amongst the different communities within the municipality. Thus, listing the main water resources as taps, boreholes, rivers, dams, weirs and groundwater as the major water resources in the VDM. It is important to establish the source of water within the Vhembe district, since the municipality exists in an area that is predominantly rural. As such determining the source, should assist in establishing the extent to which the municipality will be obligated to provide this much needed resource. The challenge presented here is that since the municipality is lagging with the provision of water services through provision of potable water via piped infrastructure (water coming out of taps), community members are finding alternative means to source water. Often, they use untreated, contaminated, and unsafe water, which may lead to numerous illnesses, such as diarrhoea and cholera.

*8.3. Vhembe District Municipality Water Service Provision Strategy*

During the interview, the participants indicated that the district has a water service provision strategy, which needs to be reviewed periodically to ensure alignment with other sector strategies such as the community development strategy. The district should continue to review the strategy and realise its objectives to improve the existing water distribution to the local municipalities. The participants also highlighted that the DWS, in conjunction with VDM, has developed a water supply reconciliation strategy that focuses on current water use, water requirements and availability, and existing water supply infrastructure in various clusters within the Vhembe district. The strategy aims to measure the level of water service delivery offered by the district municipality in various clusters and assess the existing water supply infrastructure. Thus, the challenge associated with the use of outdated water service provision strategy is that the information does not assist in ensuring that the number of new emerging communities are matched against the existing plans to deliver water resources in the municipality.

*8.4. The Impact of Poor Water Service Delivery to Communities within the Various Local Municipalities in the VDM*

The study participants indicated that poor water service delivery in Vhembe district is affecting the daily livelihoods of people from the local communities, with some having to collect water from neighbouring villages. Tswinga village, for example, has the challenge of accessing piped water daily, resulting in the residents having to collect water from neighbouring villages such as Muledane, Maniini and Shayandima. Some of the participants indicated that illegal water connections are the result of poor water service to communities in the Thulamela and Mutale local municipalities. Therefore, poor water service delivery compromises the quality of life of the concerned community members.

*8.5. The VDM Has the Competence to Deliver Good Quality Water Services to Local Communities*

There was a view by some of the study participants where it was stated that that Vhembe District has the competence to supply good quality water services; however, the challenge lies in the variance in the level of water supply to the communities. Study participants also revealed that the district is understaffed, which affects service delivery, particularly the performance within the technical water services division. Furthermore, financial constraints have made it difficult for the municipality to implement the major water projects in some of the rural areas including Ntshabalala and Mulima.

### 8.6. Cooperative Government Challenges in the Vhembe District Municipality

This section provides an explanation of the investigation on the cooperative government challenges in addressing water service delivery in the VDM. To report on the challenges of cooperative government, the research used interviews to gather information from the participants. According to the interview guides, the second part of the interview focusses on the cooperative government challenges in the delivery of water services in the district. The provision of water services in municipalities depends on the effective collaboration with the other two spheres of government. Thus, through the principle of cooperative government, the national sphere would develop policies and municipalities would be the implementing agents of such policies. Therefore, it is important that municipalities, as they implement and facilitate the delivery of such services, must consider capacity and capability. This implies that municipalities should consider their own context towards the implementation of services.

### 8.7. Implementation of Water Service Delivery Strategy

During the interviews, study participants mentioned the need for collaboration with stakeholders such as the South African National Civics Organisation (SANCO) and other civic organisations, as well as universities and community forums. Participants were of the view that this would strengthen the mechanisms of cooperative government in the district as the municipality and its stakeholders would speak with one voice on how to improve water service delivery to communities. In addition, participants noted that the integration of various plans in the district could contribute to an improvement in the strategy for water service delivery. For example, the water service delivery plans of the local municipalities should align with that of the district to promote mutual relationships and strengthen cooperative government.

### 8.8. Effective Coordination and Integration of Stakeholders

The study participants revealed that the VDM has an efficient water service delivery plan. Various stakeholders are involved in the planning process and assigned roles and responsibilities. The water plan identifies the risks that are associated to water service delivery, and stakeholders such as water boards and community forums are able to address these risks during the planning process. The water service delivery plan assists in identifying the required action and costs that will be incurred and determining the level of water service delivery to the local municipalities.

### 8.9. Municipal Capacity

Participants indicated that municipal capacity is a major challenge affecting cooperative government in the Vhembe district. The participants emphasised that the VDM has limited capacity since it is understaffed in the technical services department, and this affects the distribution of water supply to the local communities. Participants indicated that the district should employ artisans, engineers, and plumbers in order to improve water infrastructure development in the district.

## 9. Conclusions and Recommendations

This article illustrates that the South African government has developed several plans, strategies, policies and frameworks on how the water sector should manage its resources, including the distribution of water supply to the local communities. One of the most significant challenges facing the water sector, however, is implementation. South Africa is known for developing some of the best strategies, policies and legislative framework; however, there seems to be a lack of capacity and political will to implement. This article addressed the nature of the practice of cooperative government in the context of water service provision within the three-sphere system. Furthermore, this article elaborated extensively on the degree to which cooperative government is an effective tool for water provision by district municipalities.

The possible strategies that could be used to overcome challenges of water service delivery in the Vhembe District Municipality are summarised as follows. These strategies are motivated by the country's needs to adopt holistic kind of approaches in addressing implementation challenges. A similar approach that could be possibly enhanced is the District Development Model, which seek to align the developmental planning within municipalities with the budgeting and finances that are usually allocated by the national and provincial governments. To adequately implement the already existing policy framework, we suggest that the Vhembe District Municipality considers the following:

(i) Establish and strengthen its technical water service satellite offices in the respective local municipalities. This would assist with the decentralisation and devolution of service provision, thereby ensuring that the water service strategy and plan are easily implemented by the local municipalities with the district municipality playing an oversight role.

(ii) Improve the mechanism of cooperative government to allow for the strengthening of collaboration between the technical units of the municipality and external stakeholders. Through such an engagement, it will be possible for the municipality to explore alternative service delivery approaches, such as through Public Private Partnerships.

(iii) Manage talent and prioritise succession planning in departments, units and sections that require critical skills. Most employees working in the water technical services have a scarce skill and as such, due to the instability of municipalities, would often leave the employ of the municipality in favour of the private sector. This creates a huge gap in terms of the requisite skill in within the municipality, which leads to slow response to infrastructural and other water related problems in communities.

(iv) Properly align its established water service plans and strategies within its Integrated Development Plan. This will foster the much-needed coordination and integration of the service within the broader plans of the municipality.

**Author Contributions:** Conceptualization, A.K., O.S.M. and M.M.T.; methodology, A.K., O.S.M. and M.M.T.; writing—original draft preparation, A.K. and O.S.M.; writing—review and editing, O.S.M. and M.M.T. All authors have read and agreed to the published version of the manuscript.

**Funding:** This research received no external funding.

**Institutional Review Board Statement:** The study was conducted in accordance with the Declaration of Helsinki, and approved by the Ethics Committee of the Faculty of Economic and Management Sciences at the University of Pretoria on the 11 November 2020.

**Informed Consent Statement:** Informed consent was obtained from all subjects involved in the study.

**Data Availability Statement:** Not applicable.

**Conflicts of Interest:** The authors declare no conflict of interest.

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
