# Peer review of "Complexities of Intergovernmental Relations in Water Service Provision: A Developmental Local Government Perspective"

_socsci, doi:10.3390/socsci12110614_

Round 1
Reviewer 1 Report
Comments and Suggestions for Authors
This paper provides a summary of the cooperation of the government in water management in South Africa. The study utilized qualitative research methods, including interviews and questionnaires. One of the primary objectives was to understand the complexities of water management and apply this understanding to water service delivery strategies and plans.
One major concern I have pertains to the methodology employed. The paper mentions the use of qualitative research methods, specifically interviews and questionnaires. However, further elaboration on these methods would be beneficial, such as the number and demographic details of the samples, the sampling methodology, and how the data's reliability was ensured through these methods. It is crucial to disclose the data sources to validate the results. While the paper refers to purposive sampling of experts within the water service value chain in the Vhembe District Municipality, it lacks details regarding their specific roles and durations of work, which could potentially impact the results. Additionally, clarity on whether the interviews were anonymous would be valuable. Confirming the utilization of at least fifty percent of the identified experts after the interviews is important for transparency. The paper briefly mentions secondary data in Chapter 2; providing insights into the nature of this data and its usage would enhance comprehension.
In Chapter 3, it would be beneficial to evaluate the significance of each section and establish clear connections to the preceding chapters.
Regarding Chapter 6, the authors state, "In the previous sections, light was shed on the legal framework and institutional arrangements that guide the provision of water services in South Africa." It raises the question of whether "previous sections" refer to Chapters 1-5 and if they contain existing data. To avoid confusion and bias, I recommend separating the presentation of existing situations from the results of the interviews.
In Chapter 6, "Discussion and Findings," which should be Chapter 7, the contents primarily encompass findings from the interviews or questionnaires. Including discussions supported by relevant theories or exemplary cases from other areas would enrich this section. Additionally, specifying the number of responses, such as "30 out of 50 samples revealed...," would enhance the persuasiveness of the arguments for the readers.
Chapter 8, "Conclusion and Recommendations," lacks supportive reasons or justifications for the proposed solutions. Providing reasons behind the proposed solutions would help readers comprehend the situation better and align these proposals with their own ideas.
Furthermore, one of the study's goals is to analyze the application of the principle of cooperative government as an effective tool to ensure water service delivery. However, this aspect is not analyzed in the paper.
Minor issues such as typos and inconsistencies in citation style should also be addressed to adhere to standard publication guidelines.
Author Response
Reviewer 1:
- In Chapter 3, recommended that we evaluate the significance of each section and establish clear connections to the preceding chapters. A paragraph has been added to effect the recommendation.
- Recommended the separation of the presentation of existing situations from the results of the interviews. Chapter 6 is now dedicated to the current state of affairs whereas chapter 7 presents the results of interviews.
- In Chapter 8 we have provided reasons behind the proposed solutions.
- On line 436 we have added an explanation on how the application of the principle of cooperative government as an effective tool to ensure water service delivery in VDM.
- Typos and inconsistencies in citation style have been addressed.
Reviewer 2 Report
Comments and Suggestions for Authors
1. In order to explore the practice of cooperative government among the three spheres of government in the provision of water services to communities in South Africa, this study collects and analyzes data from interviews, literature, reports and document of governments.
2. The content is not succinctly described and contextualized with respect to previous and present theoretical background on the topic.
3. Cited references does not strongly relevant to the topic.
4. The content of discussion could be more broadly and deeply to explore the aim of topic.
5. I suggest replacing the word of “DWS” on line 117 by “Department of Water and Sanitation (DWS)”.
6. For the same reason, please replace “Department of Water and Sanitation (DWS)” on line 223 by “DWS”.
Author Response
Reviewer 2:
- The content is not succinctly described and contextualized with respect to previous and present theoretical background on the topic. Revisions made ensured that the article is presented in a more logical manner.
- More relevant literature on intergovernmental relations and water service(s)/delivery was consulted and utilised, to ensure that the literature remains strongly relevant to the topic.
- The content of discussion could be more broadly and deeply to explore the aim of topic. The content in the discussion and findings section has been revised to deeply reflect on the main aim of the topic.
- The word of “DWS” has been replaced by “Department of Water and Sanitation (DWS)” on line 117.
- The “Department of Water and Sanitation (DWS)” has been replaced by “DWS” on line 224.
Reviewer 3 Report
Comments and Suggestions for Authors
Dear Authors
The article analyzes the complexity of intergovernmental relations in the provision of water services in the Vhembe District Municipality in South Africa. The authors focus mainly on the analysis of the legal provisions in force in the country. They are supplemented by information obtained through interviews with employees involved in water distribution in this area. In my opinion, the research procedure was not properly planned. The goals that the authors try to achieve appear several times in the work, and they should be clearly defined in the introduction. After reading the text, it can be concluded that the answers to the research questions are found in the relevant legal provisions in force in South Africa. Moreover, the first goal is not clearly defined: what are the three zones? line 39-40. The article lacks information about what questions were asked during the interview. It is difficult to assess the credibility of the research conducted. No information whether problems with access to water are common? How many people don't have access to water? The presented conclusions are very general. In the first part, they were borrowed from the literature (Fjelstad (2004); Muller et al. (2009), and should rather be the result of conducted research. Moreover, conclusion ii) line 482 and iii) line 485 are valid everywhere in the world and do not refer to the analyzes performed. The article does not contribute anything new to the literature and is extremely local in nature.
Author Response
Reviewer 3:
- The three spheres of government (i.e. national, provincial and local) have been added on line 52.
- The goals that the authors try to achieve appear several times in the work, and they should be clearly defined in the introduction. The repetition of the aim of the paper has been streamlined and covered in the introduction along with the questions that paper seeks to address.
- A new section has been introduced (i.e. current state of affairs) to highlight the problem of lack of access to water. Thus, indicating the magnitude of this problem specifically in a developing country like South Africa.
- The presented conclusions are very general. In the first part, they were borrowed from the literature (Fjelstad (2004); Muller et al. (2009), and should rather be the result of conducted research. Moreover, conclusion ii) line 482 and iii) line 485 are valid everywhere in the world and do not refer to the analyzes performed. The conclusions and recommendations have been revised to reflect directly what has been uncovered in the research.
- The article does not contribute anything new to the literature and is extremely local in nature. The paper endeavours to contribute to the understanding of a global issue but from a local perspective. Additional information has been added to substantiate the importance of thinking from a global point of view but acting from a local perspective. The case of South Africa is unique given its historical background hence some of the recommendations are specific and local in nature.
- The types of questions asked to participants and the category of the participants haves been discussed and elaborated in the methodology section (see line 96 -116)
Reviewer 4 Report
Comments and Suggestions for Authors
Complexities of intergovernmental relations in water service provision: a developmental local government perspective
The paper addresses the key issue of that institutions create boundaries, and thus organisations are bounded, but systems do not necessarily have the same boundaries either with the organisations or other systems. Hence, a critical question is how effectively organisations can form and maintain the necessary bridging mechanisms to co-act. The study area is especially interesting because of the difficult and expensive problems of providing water services in Low Income Countries and because South Africa an articulated theory of co-operative inter-government relationships unlike many countries where rather ad hoc essentially hierarchical arrangements still exist. So I looked forward to reading this paper but came away somewhat disappointed. I should say that I read this from the perspective of someone concerned with the delivery of water services rather than from a sociological perspective.
Most of the paper is taken up describing the legislative and administrative arrangements in South Africa (line 376 to end at L493); and I was waiting to see what were the problems on the ground in Vhembe District, which organisations had to be involved and what were the roles of each, and how were they solved? I did not get this; the paper gives the problems as resource and skill shortages but does not say anything specifically about co-operation; although it mentions some issues (L229) without detail. Two questions I hoped that I get information were on my suppositions that the particular nature of a water provisioning system has to be reflected in the institutional arrangements. Thus, that what works for borehole and pump will not be sufficient for dam and piped distribution system. I didn’t find anything on the nature or structure of the system involved but just an all inclusive descriptive listing (L395). A crude functional outline of water supply system would be:
Extraction > transfer > treat > distribute
In turn, there are sometimes different organisations responsible for different functions; in California, the first two are separated from the second two in some cases; the Water Board is responsible for the first two in parts of Vhembe.
https://www.sabcnews.com/sabcnews/vhembe-district-municipality-needs-r4-billion-to-refurbish-water-infrastructure/
Or, because of the physical and often skill economies of scale, horizontal co-operation is adopted eg the Syndicats of municipalities in parts of France and Belgium where several municipalities form a co-operative to deliver water services.
It would be helpful to have a table defining which organisation is responsible for providing and/or operating and maintaining each functional component of the water supply system although this may vary according to, for instance, whether the extraction is via a borehole or from a reservoir.
Here, some basic data on the Municipality would have been helpful; that it has a population over 1.1 million and covers an areas of over 21,000 km2 has implications for the water supply problem.
Secondly, that all governance is done by people and hence inter-personal skills are important so I would have liked to have heard something about what made for successful interactions leading to co-action. What lessons did the interviewees have for other areas?
Is there more to be gained from the qualitative material?
So, overall, I consider the paper to be acceptable with a few minor additions but disappointing as it stands and I hope that the author(s?) bring out some lessons for others from the case study.
Typos and things:
L38 What are the three spheres of government? Inferred later but I had to look up it
https://etu.org.za/toolbox/docs/govern/spheres.html
L438 SANCO – say what this is
L472 Muller not in references
Author Response
Reviewer 4:
- I was waiting to see what were the problems on the ground in Vhembe District, which organisations had to be involved and what were the roles of each, and how were they solved? Additional information has been added to substantiate the problems faced by Vhembe District on lines 44 to 70.
- I didn’t find anything on the nature or structure of the system involved. The focus of the article was on the intergovernmental relations and cooperative government in the context of water service provision. Thus, the three-sphere system was articulated, in particular new information was presented to describe the structure of local government in South Africa and how it plays an important role in water service provision.
- Three spheres of government have been added on L52
- Full words added for the acronym SANCO “the South African National Civics Organisation (SANCO)” on L528.
- References to sources in the conclusion i.e. Muller et al.(2009) and Fjelstad (2004) have been removed.
Round 2
Reviewer 2 Report
Comments and Suggestions for Authors
This paper has revised. I have no more comment.
Reviewer 3 Report
Comments and Suggestions for Authors
Dear Authors
Thank you for the opportunity to review the manuscript again.
The authors considered the suggested corrections, which is why the article is better. I am inclined to publish the article in its current version, leaving the final decision to the Editors.